# Is all greenspace created equal? Assessing the relationship of public parks and greenness on academic achievement in Washington state

Lea Fortmann[ID]*, Isha Rajbhandari

Department of Economics, University of Puget Sound, Tacoma, Washington, United States of America

* lfortmann@pugetsound.edu

## Abstract

In educational environments, proximity to greenspace can increase concentration, attentiveness, self-discipline, improve classroom engagement, and support cognitive development. This study contributes to this growing literature by examining the potential benefits associated with public parks on academic performance across different grade levels in four counties in western Washington state, USA. We use a five-year panel data set to assess the impact of a school's proximity to a public park on the percentage of students in each grade who meet or exceed proficiency standards for statewide assessment tests while controlling for general greenness, along with a number of socioeconomic and demographic factors. Our results suggest that having a public park within 800 meters of the school is associated with an increase in the percentage of students that meet or exceed the standard for language arts and math standardized tests for middle schoolers. The local greenness of the school grounds and surrounding area, measured by NDVI, had no impact on test scores. This suggests that the parks themselves could have a positive relationship, beyond increased exposure to greenness, and potentially provide academic benefits to students. We do not find a significant impact of a school's proximity to public parks for elementary or high school students, which we hypothesize could be for various reasons. Though the literature on the benefits associated with public parks is well established, this is the first study, to our knowledge, to investigate their potential relationship with educational performance.

## Introduction

Access and proximity to public parks and greenspaces have been shown to provide a number of physical, mental, and social benefits. Research has found that children and adolescents who live in close proximity to parks are more likely to be physically active [1,2] have lower obesity rates [3], reduced stress levels, and better emotional wellbeing [4,5]. Simply being exposed to greenspace has been shown to improve

provided the original author and source are credited.

**Data availability statement:** All relevant data are within the manuscript and its Supporting information files.

**Funding:** The author(s) received no specific funding for this work.

**Competing interests:** The authors have declared that no competing interests exist.

cognitive functioning, memory, and attention spans [5]. In addition to providing a space for physical activity, social gatherings, and interactions with nature, parks and greenspaces also improve other aspects of quality of life through air and water purification, microclimate stabilization, and noise reduction [6].

While the body of research on the associated benefits of greenspace and interactions with nature is well established, a subset of emerging research is focused on the relationships of greenness and greenspaces on academic performance, though the findings are wide ranging and, at times, mixed. Wu et al. (2014) looked at the surrounding greenness of elementary schools in Massachusetts, measured by the normalized difference vegetation index (NDVI), and found a positive relationship between NDVI in the spring and school-wide academic performance for third graders [7]. Leung et al. (2019) expanded on this research to include more grade levels (grades 3–10) in Massachusetts and also found a positive association with the greenness of a school and the percentage of students who score proficient or higher on standardized tests across all grades [8]. Other studies had similar results in Minnesota [9], Michigan [10], and Washington D.C., where researchers found that schools with more trees had better performance on standardized test scores [11]. In Washington state, a study by Kuo et al. (2021) compared general greenness and tree canopy and found that both predicted higher standardized test scores, though tree canopy was a stronger predictor than overall greenness measured by NDVI [12]. These findings also hold internationally, where researchers in Australia examined the relationship of school vegetation on academic performance for primary school children (grades 3 and 5) and found statistically significant, positive associations with standardized test scores in reading and mathematics [13,14]. In the United Kingdom, researchers found that 11-year-olds living in greener urban neighborhoods had better spatial working memory [15], which is a strong correlate of academic performance [16].

On the other hand, a review of the literature on greenspace and its relationship with school performance by Browning and Rigolon (2019) looked at 13 peer reviewed studies focused on greenness surrounding school campuses and found mixed results, with the majority of studies not being statistically significant [17]. Additionally, a replication of the Wu et al. (2014) study by Browning et al. (2018) in Chicago found a near zero, but negative statistically significant relationship between NDVI and school performance for students in grade 3, though the authors note that this is a less green and more underprivileged neighborhood than the original Massachusetts study area [18]. Likewise, Hodson and Sander (2019) looked at a sampling of urban schools across the United States to see if different levels of vegetative cover impact reading and math proficiency at the high school level and found that tree cover did not have a significant impact, though non-forest vegetation (e.g., shrubs and grassland) had varying effects on academic performance depending on urban density [9]. Internationally, a study of German adolescents found mixed results and concluded that there was not an association of increased greenspace and improved academic performance [19].

Beyond general greenness, another strand of research examines how interactions with and time spent in nature could benefit school performance. In a study by Wallner

et al. (2019), high school students were given a one-hour lunch break in a small park, a large park, or a forest [20]. Measures of wellbeing and concentration were taken before and after, and the results found that spending time in nature improved student performance across all measures, with the strongest impacts for students who spent time in the larger parks and forests. Another study looked specifically at the potential for parks and structured recreation to lessen declines in self-regulation and self-efficacy among low-income middle school students in Miami, Florida [21]. They found promising results, suggesting that participation in parks-based afterschool programs mitigated increases in anxiety and depression for student participants at a critical stage of adolescence. Finally, a review of 14 different studies on short-term exposures to nature and academic performance concluded that exposure to nature in the middle of the school day, lasting anywhere from 10 to 90 minutes, was positively associated with cognitive functioning through attention restoration and reduced mental fatigue across a range of grade levels, from elementary school to university [22].

This paper aims to bring together both strands of literature and goes beyond measures of general greenness to examine the potential educational co-benefits correlated with accessibility to public parks in Washington state, USA. While there is extensive literature on benefits associated with access and proximity to greenspace, with an increasing focus on the benefits of greenspace and education, to our knowledge, this is the first study that has made a connection between educational outcomes and a school's proximity to a public park.

## Materials and methods

### Study area

The study area covers four contiguous counties in western Washington state (King, Pierce, Thurston, and Snohomish) along the I-5 corridor (Fig 1). We focus on these four counties due to their urban nature rather than the more rural areas of eastern Washington. Additionally, the western side of the state is more homogeneous in terms of year-round greenness, given its temperate climate, compared to the eastern side of the state which has a drier, semi-arid climate.

### School data

This study uses publicly available school and grade level data from the Washington Office of Superintendent of Public Instruction (OSPI) [23]. We compiled the data into a five-year panel dataset of 825 schools for the academic years 2014–2015–2018–2019 at the elementary, middle, and high school levels. We also include a number of covariates in our analysis at the grade level and school level to control for student demographics and school quality, including: race, gender, income, and teacher quality variables. To account for schools located in more rural areas in western Washington, we include data from the National Center for Educational Statistics (NCES) that classifies schools as rural based on their distance from an urban area [24].

Table 1 shows the descriptive statistics of the school-level variables broken out by school type. Elementary schools generally include grades 1–6, middle schools include grades 7 and 8, and high schools include grades 9–12, though the exact grades in any school may vary from school to school. In our sample 30–40 percent of middle schools also include grade 6 and a smaller number of middle schools include grade 9. For the academic year 2018–2019, our total sample across the four counties includes 444 elementary schools, 263 middle schools, and 118 high schools. On average, 59 percent of students met or exceeded the standard for state assessment tests in elementary school, 57 percent in middle school, and 43 percent in high school. At least 40 percent of the students in each grade were identified as being low-income, which is based on their eligibility for free and reduced-price meals at school. The average racial composition was relatively consistent across elementary, middle, and high schools with approximately 50 percent or more of the students being white. On average, teachers had approximately 12 years of teaching experience and more than 60 percent of them had a Master's degree. Our main independent variable of interest is proximity to a public park, which is based on a school having a public park within an 800 meter radius. In our sample, 66 percent of elementary schools, 60 percent of middle

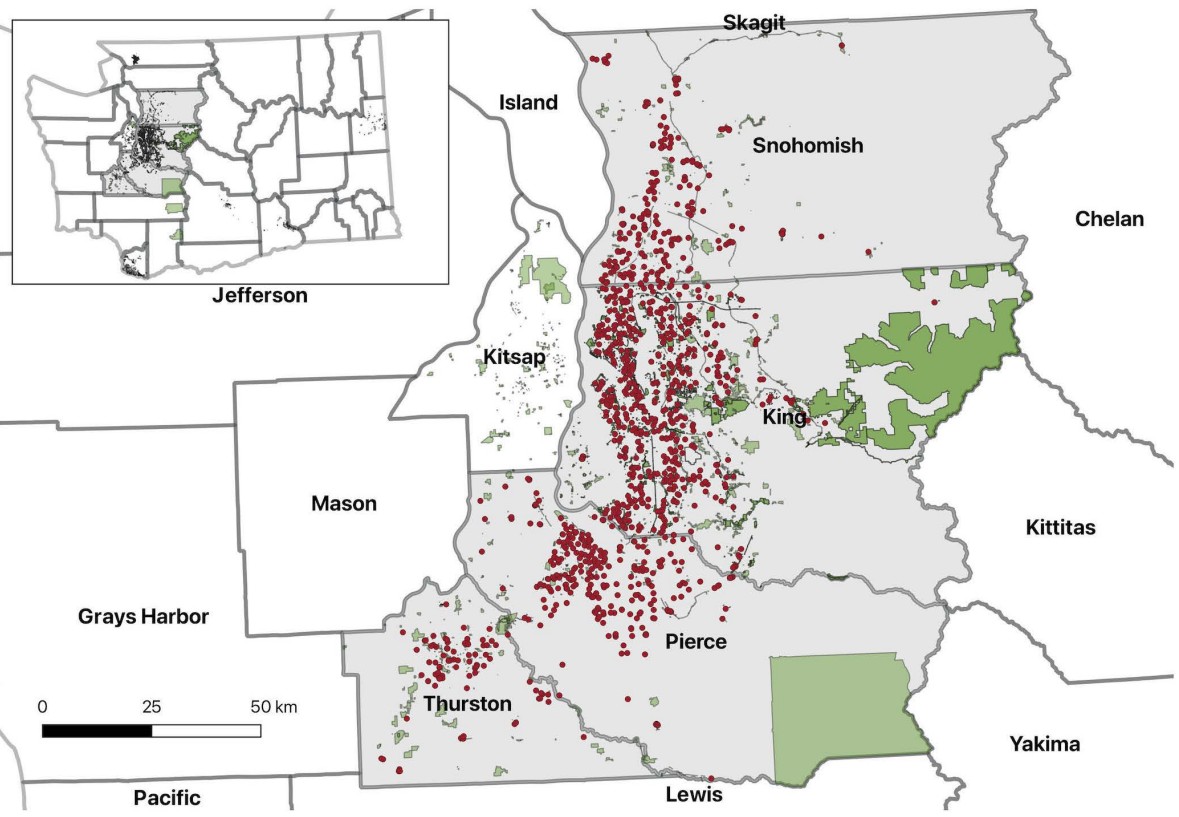

**Fig 1. Map of Study Area in Washington State.** The green areas on the map represent public parks and the red dots represent the K-12 schools located in the four counties being studied. Republished from the Washington Office of Superintendent of Public Instruction, Washington State Parks, and Washington Department of Natural Resources under a CC BY license.

**Table 1. Descriptive statistics for elementary, middle, and high school variables.**

| PANEL A: Variable | Elementary School | | Middle School | | High School | |
|---|---|---|---|---|---|---|
| No. of schools (2019) | 444 | | 263 | | 118 | |
| No. of schools with a park within 800m (2019) | 292 | | 157 | | 79 | |
| No. of schools in Urban areas (2019) | 380 | | 213 | | 94 | |
| Panel B: Variable | Mean | SD | Mean | SD | Mean | SD |
| Percent met or exceeded standard on test | 58.9 | 18.51 | 56.82 | 18.76 | 43.16 | 29.06 |
| Percent low-income | 44.33 | 26.93 | 40.83 | 25.11 | 40.83 | 25.13 |
| Percent female | 47.86 | 10.07 | 47.1 | 12.64 | 47.58 | 13.87 |
| Percent white | 49.84 | 23.3 | 52.64 | 23.94 | 53.86 | 24.64 |
| Percent Native Indian | 0.91 | 4.54 | 1.23 | 5.94 | 1.59 | 6.78 |
| Percent Asian | 10.98 | 11.75 | 10.57 | 11.26 | 9.89 | 11.06 |
| Percent African American | 7.66 | 10.43 | 7.1 | 9.91 | 7.83 | 11.5 |
| Percent Hispanic | 18.17 | 13.15 | 16.27 | 11.66 | 16.00 | 11.7 |
| Percent Hawaiian | 1.55 | 2.99 | 1.47 | 3.23 | 1.39 | 2.57 |
| Teacher average years of experience | 12.19 | 3.41 | 12.75 | 3.9 | 12.31 | 4.8 |
| Percent teachers with Master's degree | 62.59 | 13.19 | 66.74 | 14.62 | 65.37 | 18.49 |

The data comes from the Washington Office of Superintendent of Public Instruction (OSPI) [23]. The values are averaged across grades for each school level for the years 2014–2019 except when noted.

 

schools, and 67 percent of high schools had a park within an 800 m radius for the academic year 2018–2019. While the exact number of schools vary slightly from year to year, they are generally consistent for the period of analysis.

## Parks and greenspace data

The park data used for the spatial analysis come from county and city parks and recreation department websites [25–28]. The data was cleaned to remove areas such as golf courses, cemeteries, dog parks, etc. that are not publicly accessible or would not be places where students would reasonably spend time outside of school. Places that count as a "park" include parks, beaches, playfields, and nature trails.

We also assess the overall greenness of the area surrounding the school at various buffer distances (250, 400, and 800 meters) based on the normalized difference vegetation index (NDVI), which is a common metric used to measure vegetation greenness and density. "NDVI values range from +1.0 to -1.0. Areas of barren rock, sand, or snow usually show very low NDVI values (for example, 0.1 or less). Sparse vegetation such as shrubs and grasslands or senescing crops may result in moderate NDVI values (approximately 0.2 to 0.5). High NDVI values (approximately 0.6 to 0.9) correspond to dense vegetation such as that found in temperate and tropical forests or crops at their peak growth stage." [29].

NDVI has been used in a number of studies assessing the relationship between greenness and academic performance [7,8,12,18,19,30]. Data for the NDVI measure comes from the U.S. Geological Survey (USGS) website using Landsat 8 images of the study region. Images were selected based on the aerial photo with the least cloud cover during the months of October through December for the "fall" NDVI variable and during the months of March through May for the "spring" NDVI variable for each year in the panel dataset. Table 2 shows the mean NDVI for fall and spring for buffer areas around the schools at 250, 400, and 800 meters. The relatively small magnitude of the standard deviations, particularly in the spring, suggest that there is not substantial variation in greenness among schools, as measured by NDVI.

## Empirical methods

We use an Ordinary Least Square (OLS) fixed effects model to assess the correlation of park proximity and academic achievement in Washington state. The main outcome variables are measured as percentages and are the combined percent of students in each grade who met the proficiency standard and the percent of students who exceeded the standard across various assessment tests in each year of the study. The subjects for the assessment tests vary based on the grade-level being considered. For example, grades 3, 4, 6, and 7 take the standardized Language Arts and Math assessment tests, whereas grades 5, 8, 10, and 11 take an additional standardized Science assessment test. Our analysis does not include test results from grades 1, 2, 9, and 12 since students in these grades only take the standardized English language assessment test, which is for non-native English speakers. The main independent variable of interest is a binary variable indicating if a public park was within an 800 meter radius of the school. The 800 m radius was utilized

**Table 2. NDVI summary statistics by school level.**

| Variable | Elementary School | | Middle School | | High School | |
|---|---|---|---|---|---|---|
| | Mean | SD | Mean | SD | Mean | SD |
| NDVI fall 250 meters | 0.11 | 0.04 | 0.11 | 0.05 | 0.09 | 0.04 |
| NDVI fall 400 meters | 0.1 | 0.04 | 0.11 | 0.04 | 0.09 | 0.04 |
| NDVI fall 800 meters | 0.1 | 0.04 | 0.1 | 0.04 | 0.09 | 0.04 |
| NDVI spring 250 meters | 0.28 | 0.06 | 0.27 | 0.06 | 0.24 | 0.06 |
| NDVI spring 400 meters | 0.28 | 0.06 | 0.28 | 0.06 | 0.26 | 0.07 |
| NDVI spring 800 meters | 0.26 | 0.07 | 0.26 | 0.07 | 0.24 | 0.08 |

The values are based off of all five school years for each season and buffer area.

to ensure that we are accounting for public parks within walking distance from the school [31, 32]. We also consider a number of other park-related variables indicating if a park is within 250 m or 400 m of the school, distance to the nearest park (in meters), and the percentage of park area within an 800 m buffer area around the school (see Table 4). The main control for general greenness is the NDVI which allows us to examine the relationship with parks on educational outcomes beyond their more general contribution to increased greenness surrounding the school. We use NDVI based on an 800 m buffer area in the spring in the main model to parallel the park radius measure, but we also examine NDVI in the fall with various buffer distances.

## Covariates

We include a number of economic and demographic covariates at the grade, school, and school district levels. At the grade-level, we control for gender, race, and the percent of low-income students for each year. At the school level, we control for school size based the total number of enrolled students, and teacher quality based on the average number of years of experience and the percent of teachers with a Master's degree on an annual basis. To control for neighborhood characteristics around the school that could also be related to park placement and student test scores, we include a variable for mean household income within an 800 m buffer zone. This is based on the weighted average of income at the census tract-level from the American Community Survey five-year estimate from the years 2017–2021. We also include a binary variable for whether the school is located in an urban or rural school district since it can have important implications for the students' access to greenspace. At the school district level, we control for annual per pupil funding and property taxes (logged values) which are the primary source of school funding in Washington state. These variables account for the amount of money a school district has in its budget which affects the resources, teachers, facilities, and support a school can provide to its students and serve as additional covariates for school quality.

Finally, we add county-level and school district-level fixed effects to account for any county- and school district-level time-invariant factors such as differences in regulations or policies that vary across districts that could influence educational outcomes. County and school district fixed effects also control for spatial factors that might impact academic achievement across districts [7]. Year fixed effects are added to control for any time-variant trends such as changes to state educational policies or broader economic shifts across western Washington that might impact students' academic performance. Standard errors are clustered at the school-district level.

## Results

Overall, our results suggest that in contrast to previous studies [7–10] the presence of greenness alone, measured by NDVI, is not correlated with better academic outcomes. However, we do find that having a public park within a half mile radius (800 meters) of a school has a positive and statistically significant relationship with the percentage of students who met or exceeded standards for assessment tests for middle schools, but there is not a discernible impact at the elementary or high school levels (see Table 3). Specifically, the results indicate that the presence of a public park within the 800m buffer area of the school is associated with a 4.2 percent increase in the number of students who met or exceeded proficiency standards on the assessment tests for middle schools on average, while the results for elementary and high schools are much smaller in magnitude and not statistically significant. Notably, the greenness of the school grounds and surrounding area within an 800 meter buffer, accounted for by the NDVI variable, is positive but not statistically significant for any school level.

Additional results show that the percent of low-income students at the school has a statistically significant negative relationship with educational outcomes across all three school levels. This suggests that grades with a higher percentage of low-income students experience a significant reduction in the percentage of those who meet or exceed proficiency standards on assessment tests, which is consistent with previous literature examining the relationship between academic achievement and socioeconomic status [33]. The mean household income of the neighborhood immediately surrounding

**Table 3. Regression results for all school levels.**

| Outcome variable: Percent met standard | Elementary | | Middle School | | High School | |
|---|---|---|---|---|---|---|
| | Coef. | SE | Coef. | SE | Coef. | SE |
| Park within 800 m | 0.47 | 0.55 | 4.17*** | 1.62 | −0.20 | 2.06 |
| NDVI (800 m spring buffer) | 6.68 | 9.42 | 9.94 | 10.01 | 1.61 | 22.7 |
| Mean household income (800 m buffer) | 0.004** | 0.002 | 0.002 | 0.002 | 0.001 | .004 |
| Percent low-income students | −0.33*** | .036 | −0.34*** | 0.04 | −0.29*** | 0.07 |
| Percent female students | .017 | 0.03 | 0.09 | 0.07 | −0.01 | 0.07 |
| Percent white | −0.05 | 0.04 | 0.07 | 0.07 | −0.32** | 0.14 |
| Percent Native Indian | −0.31* | 0.16 | −0.50** | 0.21 | −0.47*** | 0.15 |
| Percent Asian | 0.16*** | 0.05 | 0.34*** | 0.08 | 0.153 | 0.12 |
| Percent African American | −0.22*** | 0.06 | −0.14 | 0.12 | −0.48*** | 0.16 |
| Percent Hispanic | −0.17*** | 0.06 | −0.004 | 0.11 | −0.50*** | 0.13 |
| Percent Hawaiian | −0.19 | 0.12 | −0.56** | 0.28 | −0.58** | 0.24 |
| Teacher average years of experience | 0.78*** | 0.22 | 0.07 | 0.22 | 0.07 | 0.32 |
| Percent teachers with Master's degree | −0.05 | 0.04 | 0.16*** | 0.04 | 0.25*** | 0.08 |
| Log general per pupil funding (district) | −0.14* | 0.07 | 0.55*** | 0.13 | −1.14*** | 0.41 |
| Log school district property tax (district) | −4.64* | 3.21 | 3.15 | 5.49 | 19.82*** | 6.08 |
| Rural dummy | 0.12 | 1.96 | −078 | 1.43 | 6.76** | 2.90 |
| Total students | 0.01** | 0.005 | 0.01*** | 0.003 | 0.007*** | 0.02 |
| District, county, and time fixed effects | Y | | Y | | Y | |
| Adjusted R-squared | 0.59 | | 0.60 | | 0.25 | |
| Observations | 21,321 | | 6,256 | | 2,434 | |

The results reported here are based on all grades within each school level that took the math, science, and/or language arts assessment tests across all five years of analysis. Results from the English language assessment test are excluded from the regression results. ***p < 0.01, **p < 0.05, and *p < 0.1.

the school, however, does not seem to be related to the school's academic performance. Racial and gender compositions vary in their significance and magnitude across school levels. Generally speaking, the percentage of Asians tends to be associated with an increasing percentage of students that met or exceeded test standards, while the percentages of Black, Hispanic, and Native American students generally have a negative association across all school levels, which is also consistent with previous literature [34]. In terms of teacher quality, having a Master's degree is positive and statistically significant at the middle and high school levels, while more teaching experience is statistically significant only at the elementary school level. The results for per pupil funding and property tax variables vary across school types and are not consistently positive or negative. This could be due to different budgetary models and funding priorities across school districts.

## Middle school results

Given that the results in Table 3 depict a significant relationship between public parks and standardized test scores at the middle school level, we provide a more in-depth analysis examining separate grade and test level effects (see Table 4). Overall, we find that the presence of a public park within 800 meters of a middle school is associated with more than a 5 percent increase on the percentage of students who meet or exceed proficiency standards in assessment tests for Language Arts, Math, and Science in grade 8. For grade 7, Language Arts and Math are both statistically significant, but the coefficient size is smaller at 3 percent. The results for grade 6 in middle school are positive but not statistically significant which may, in part, be due to the smaller sample size. Grade 6 is split between elementary and middle schools, though

**Table 4.** Regression results for middle school grade levels.

| Outcome variable: Percent met standard | Grade Six | | | | Grade Seven | | | | Grade Eight | | | | | |
|---|---|---|---|---|---|---|---|---|---|---|---|---|---|---|
| | Language Arts | | Math | | Language Arts | | Math | | Language Arts | | Math | | Science | |
| | Coef. | SE | Coef. | SE | Coef. | SE | Coef. | SE | Coef. | SE | Coef. | SE | Coef. | SE |
| Park within 800m | 1.48 | 1.74 | 1.31 | 2.04 | 3.21** | 1.44 | 3.54* | 2.07 | 5.64*** | 1.66 | 5.56*** | 1.97 | 5.71*** | 2.05 |
| NDVI Spring (800m) | −11.4 | 10.2 | −12.7 | 15.5 | 6.94 | 9.842 | 16.9 | 14.7 | 5.35 | 9.46 | 21.1 | 13.23 | 20.4 | 14.1 |
| Mean HH income | −0.03 | 0.03 | −0.01 | 0.03 | 0.04 | 0.03 | 0.03 | 0.03 | 0.01 | 0.02 | 0.02 | 0.03 | 0.02 | 0.03 |
| Percent low-income students | −0.28*** | 0.07 | −0.32*** | 0.07 | −0.35*** | 0.063 | −0.33*** | 0.06 | −0.36*** | 0.05 | −0.36*** | 0.05 | −0.31*** | 0.06 |
| Percent female students | 0.14 | 0.10 | −0.05 | 0.10\ | 0.09 | 0.116 | −0.006 | 0.09 | 0.09 | 0.09 | 0.09 | 0.07 | 0.09 | 0.11 |
| Percent White | −0.25 | 0.13 | .016 | 0.18 | 0.06 | 0.094 | 0.17* | 0.10 | −0.10 | 0.11 | 0.08 | 0.12 | 0.10 | 0.11 |
| Percent Native Indian | −0.80*** | 0.25 | −0.65* | 0.37 | −0.45* | 0.250 | −0.21 | 0.29 | −0.85*** | 0.26 | −0.57** | 0.22 | −0.25 | 0.28 |
| Percent Asian | 0.15 | 0.10 | 0.43*** | 0.17 | 0.26*** | 0.094 | .46*** | 0.11 | 0.13 | 0.11 | 0.54 *** | 0.13 | 0.30** | 0.12 |
| Percent African American | −0.40** | 0.15 | −0.09 | 0.19 | −0.06 | 0.124 | −.03 | 0.14 | −0.24 | 0.16 | −0.10 | 0.17 | −0.24 | 0.19 |
| Percent Hispanic | −0.38** | 0.15 | −0.05 | 0.22 | 0.12 | 0.126 | .05 | 0.13 | −0.13 | 0.15 | 0.05 | 0.13 | 0.004 | 0.16 |
| Percent Hawaiian | −0.51 | 0.40 | −0.50 | 0.41 | −0.62* | 0.336 | −.33 | 0.43 | −0.72** | 0.34 | −0.52* | 0.30 | −0.53 | 0.34 |
| Teacher experience (years) | −0.21 | 0.26 | 0.09 | 0.27 | 0.16 | 0.256 | .41 | 0.26 | −0.15 | 0.24 | 0.21 | 0.25 | −0.13 | 0.29 |
| Percent teachers with Master's degree | 0.11 | 0.08 | 0.05 | 0.07 | 0.15*** | 0.042 | .15*** | 0.03 | 0.15*** | 0.06 | 0.12** | 0.05 | 0.19*** | 0.06 |
| Log general per pupil fund (district) | −0.72 | 1.49 | −0.12 | 1.63 | 0.64*** | 0.174 | .88*** | 0.13 | 0.42** | 0.19 | 0.48*** | 0.14 | 0.43* | 0.23 |
| Log school district property tax | −7.38 | 6.51 | −0.39 | 8.47 | 0.93 | 6.03 | 3.29 | 6.4 | 7.91 | 9.0 | 8.45 | 7.23 | 9.72 | 8.02 |
| Rural dummy | −2.05 | 1.44 | 6.12*** | 1.54 | −2.5 | 1.72 | 2.33 | 2.0 | −4.09*** | 1.51 | −0.41 | 2.37 | −4.2* | 2.28 |
| Total students (school) | 0.005 | 0.003 | 0.004 | 0.004 | 0.008** | 0.004 | 0.01** | 0.01 | 0.001*** | 0.003 | 0.01*** | 0.004 | 0.01*** | 0.003 |
| Adjusted R-squared | 0.818 | | 0.82 | | 0.65 | | 0.69 | | 0.63 | | 0.68 | | 0.63 | |
| Observations | 622 | | 622 | | 1005 | | 1008 | | 1000 | | 999 | | 1000 | |

All models include district, county, and year fixed effects. Significance: ***p<0.01, **p<0.05, and *p<0.1.

more often it is part of an elementary school. The coefficient for NDVI is not statistically significant for any test across all grade levels.

The results in Table 4 reflect several of the more general findings shown in Table 3. The results for the racial composition variables vary across test subjects and grade levels, with Asian (positive) and Native Indian (negative) being the only variables that are consistently statistically significant. The findings suggest that socioeconomic status has a stronger association with student academic performance, where a 10 percent increase in the number of low-income students in a grade is associated with a 3 percent decrease in the number of students who meet or exceed standards. While student household income seems to matter, neighborhood characteristics seem to be less relevant given the coefficient for mean household income within 800 m of the school is near zero and not statistically significant. The results continue to show that school quality has an important relationship with educational outcomes, where the percentage of teachers with a Master's degree is positive and statistically significant for grades 7 and 8 for all test subjects, though years of teaching experience is not statistically significant in any model. Property tax revenue at the school district level, which reflects school funding and serves as a proxy for school district wealth, is also positive and significant for grades 7 and 8.

## Parks vs. NDVI results

We also examine various measures of greenspace and their association with educational outcomes to see if there is a differential relationship based on the distance and size of the park and the season and size of the buffer area for the NDVI metric. Each column in Table 5 represents a different park variable and greenness measure to assess the robustness of the results on the percentage of students who meet or exceed the standard on assessment tests in middle school with all grades and test types combined. In column (i) greenspace is measured only using NDVI, as is common in the literature. Column (ii) has the 800 m park variable and column (iii) includes both the 800 m park and the NDVI 800 m spring variables. As further robustness checks, we consider a park variable for a 400 m radius to examine differences in the proximity of the park to the school while still controlling for general greenness. We also examine variations of the park variable. Column (v) measures the percentage of park area within the 800 m buffer surrounding the school and column (vi) includes the distance to the closest park from the middle school in meters. The covariates for racial and gender composition, school quality, economic and spatial characteristics, and location and time fixed effects are the same across the six models.

**Table 5. Comparison across various measures of parks and greenspace for middle schools.**

| Outcome variable: Percent met or exceeded standard | NDVI only (i) | | Park within 800m (ii) | | Park within 800m+NDVI (iii) | | Park within 400m+NDVI (iv) | | Park percent within 800m+NDVI (v) | | Distance to park+NDVI (vi) | |
|---|---|---|---|---|---|---|---|---|---|---|---|---|
| | Coef. | SE | Coef. | SE | Coef. | SE | Coef. | SE | Coef. | SE | Coef. | SE |
| Park variable | – | – | 4.16*** | 1.59 | 4.17*** | 1.62 | 2.39* | 1.25 | 0.05 | 0.05 | −0.14 | 0.09 |
| Distance squared (m) | – | – | – | – | – | – | – | – | – | – | 0.001** | 0.0005 |
| NDVI (800m Spring) | 9.38 | 9.59 | – | – | 9.94 | 10.07 | – | – | 7.16 | 8.66 | 9.28 | 9.81 |
| NDVI (400m Spring) | – | – | – | – | – | – | 3.96 | 11.72 | – | – | – | – |
| Mean HH income (800 m) | 0.01 | 0.03 | 0.02 | 0.02 | 0.02 | 0.002 | – | – | 0.002 | 0.002 | 0.001 | 0.003 |
| Percent low-income students | −0.35*** | 0.04 | −0.34*** | 0.04 | −0.34*** | 0.04 | −0.35*** | 0.035 | −0.35*** | 0.04 | −0.36*** | 0.04 |
| Percent female students | 0.09 | 0.08 | .09 | 0.07 | 0.09 | 0.07 | 0.09 | 0.073 | 0.11 | 0.08 | 0.11 | 0.08 |
| Percent white | 0.08 | 0.08 | 0.07 | 0.07 | 0.07 | 0.07 | 0.076 | 0.077 | 0.05 | 0.07 | .036 | 0.07 |
| Percent Native Indian | −0.47** | 0.21 | −0.52** | 0.22 | −0.50** | 0.21 | −0.46** | 0.33 | −0.51** | 0.21 | −0.52** | 0.21 |
| Percent Asian | 0.34*** | 0.09 | 0.34*** | 0.08 | 0.34*** | 0.08 | 0.35*** | 0.09 | 0.31*** | 0.08 | 0.30*** | 0.08 |
| Percent African American | −0.14 | 0.11 | −0.14 | 0.12 | −0.14 | 0.12 | −0.13 | 0.11 | −0.17 | 0.11 | −0.17 | 0.11 |
| Percent Hispanic | 0.01 | 0.12 | 0.005 | 0.11 | 0.004 | 0.11 | 0.002 | 0.12 | 0.010 | 0.12 | −0.02 | 0.12 |
| Percent Hawaiian | −0.48 | 0.30 | −0.56** | 0.28 | −0.56** | 0.28 | −0.50* | 0.29 | −0.51* | 0.30 | −0.48 | 0.29 |
| Teacher experience (years) | 0.13 | 0.23 | 0.07 | 0.22 | 0.06 | 0.22 | 0.12 | 0.22 | 0.13 | 0.23 | 0.13 | 0.23 |
| Percent teachers with Master's | 0.16*** | 0.04 | 0.15*** | 0.04 | 0.16*** | 0.04 | 0.16*** | 0.04 | 0.16*** | 0.04 | 0.16*** | 0.04 |
| Log general per pupil fund | 0.55*** | 0.13 | 0.54*** | 0.13 | 0.55*** | 0.13 | 0.51*** | 0.12 | 0.53*** | 0.12 | 0.53 *** | 0.12 |
| Log school district property tax | 3.21 | 5.42 | 3.32 | 5.40 | 3.15 | 5.49 | 3.22 | 5.39 | 5.09 | 4.93 | 5.02 | 4.96 |
| Rural dummy | −0.85 | 1.98 | −0.61 | 1.45 | −0.78 | 1.43 | −0.62 | 1.69 | −0.61 | 1.98 | −0.09 | 1.87 |
| Total students (school) | 0.009*** | 0.003 | 0.01*** | 0.003 | 0.001*** | 0.003 | 0.010*** | 0.003 | 0.009*** | 0.003 | 0.009*** | 0.003 |
| Adjusted R-squared | 0.60 | | 0.60 | | 0.6 | | 0.6 | | 0.60 | | 0.60 | |
| Observations | 6,256 | | 6,256 | | 6,256 | | 6,256 | | 6,187 | | 6,187 | |

In column (v) Park variable is the percentage of park within an 800m buffer area around the school. In column (vi) the park variable is the distance to park in 100m increments with an added quadratic term for distance to park squared. All models include district, county, and year fixed effects.
***$p < 0.01$, **$p < 0.05$, and *$p < 0.1$.

As shown in columns (ii) and (iii) of Table 5, having a park within an 800 m radius of a middle school (with or without the NDVI 800 m spring covariate) has a positive and statistically significant relationship with the percent of students that meet or exceed the standard across all test types. Specifically, the coefficient value in column (ii) suggests that presence of public parks within an 800 m buffer is associated with a 4.2 percent increase in students who meet or exceed proficiency standards. The magnitude of the coefficient is similar to that of column (iii) suggesting that controlling for general greenness, based on NDVI, does not affect the relationship of the park itself on academic performance. This is further corroborated by the result in column (i) where the coefficient for the NDVI 800m spring variable alone is positive, but not statistically significant. We also reduced the size of the buffer area to see if having a public park closer to the school, within 400 meters, yields different results (column iv). In this case, the coefficient for park is statistically significant, but the magnitude of the coefficient is smaller, at 2.4 percent compared to 4.2 percent for the 800 m park variable.

When examining the size of the park, the coefficient for the percentage of park area within 800 meters of the school (column v) is not statistically significant suggesting that it is not the size of the park within the school area that matters as much as the mere proximity of having a park near the school to capture the potential benefits of parks on academic performance. Furthermore, the park percent variable only captures the area of the park within the 800 m radius of the school and does not account for the overall size of the park, which may fall outside of the buffer area. For example, a school may only show that 2 percent of the buffer area around the school contains a park, but immediately outside the buffer area students may have access to a much larger area of greenspace if the majority of the park falls outside of the buffer region. Column (vi) considers the distance to the park from the school. We include a quadratic distance term to account for a nonlinear relationship, which is statistically significant. Taken together, the distance coefficients suggest that as the distance to the nearest park increases, the percentage of students meeting the standard on assessment tests decreases on average, though at a diminishing rate. We also control for a number of other greenness measures, including NDVI in the fall and various buffer area sizes for the NDVI metric where in the extant literature NDVI is commonly measured at varying distances of 250m, 400m, and 1000m buffers surrounding the school [7,8,18]. Overall, we find that NDVI is not statistically significant for any of the various measures (see S2 Table for more details).

## Discussion

Despite some positive findings regarding the association of greenness and academic performance, the literature is ultimately mixed, with some studies finding a positive and statistically significant relationship with greenness and test scores [7–12], and others finding no relationship or even small, negative relationships [17–19]. Our results find no relationship between NDVI and school performance, but we do find a statistically significant, positive relationship between park proximity and student academic performance for middle schools. This suggests that having neighborhood parks near schools can provide more benefits than increasing greenness on school campuses. Focusing on parks, rather than just school grounds may also be more desirable since it would benefit a wider range of the population living in proximity to the park beyond just the students attending the school.

In terms of the role of parks in academic performance, this study only looks at the proximity of parks to school campuses and does not account for actual student use and engagement with the parks. Though if we assume that students are more likely to spend time in a park if it is in close proximity to their school campus, then these results are consistent with other research that has looked at the benefits associated with students' interactions with parks and natural spaces and have found positive outcomes. These studies suggest that spending time in natural habitats can lead to improvements in students' concentration, attention span, self-discipline, and classroom engagement, as well as reduce mental fatigue, stress, and depression [12,20–22,35], all of which can significantly impact academic performance reflected through student assessment test scores.

One potential factor explaining why the results are positive and significant only at the middle school level is that middle schoolers are able to be more independent and thus, could be more likely to spend time in parks in close proximity to their school whereas elementary school students are more likely to spend time in parks that are near their residence since they still need parental supervision. We hypothesize that interactions with public parks also have benefits to elementary age children's academic performance, but a more appropriate study would focus on park proximity at the neighborhood level rather than the school. On the other hand, while high school students have more independence and autonomy, they also tend to have more afterschool commitments (increased homework loads, extra-curriculars, sports, part-time jobs, etc.) and therefore have less time to spend in local parks, especially once they are able to drive and are not limited to walking or taking the bus outside of school hours. We might hypothesize that the positive impacts of parks could extend to grade 9 students in high schools, but unfortunately, this grade is not included in the analysis because the only standardized test students take in grade 9 is for their English language proficiency.

The results for middle schools are also relevant given the research surrounding adolescents, who may be uniquely positioned to benefit from the stress reducing effect of spending time in natural spaces given that they are at a critical stage of development in their life [21,36–38]. Our results here are consistent with a study by Mueller et al. (2023) that looked at the relationship between greenspace and mental health among adolescent youth in London [39]. They found that older adolescents, age 13–15, seemed to benefit more from proximity to greenspace compared to younger adolescents (ages 10–12) regarding mental health and overall well-being, where the average age of an eighth-grade student is 13 or 14 years old. On the other hand, they found some negative linkages with greenspace for younger adolescents particularly regarding negative social behaviors, hyperactivity, and inattention [39]. Another study by Feda et al. (2015) examined the benefits of proximity to parks at the neighborhood level for adolescents (ages 12–15) in Erie County, NY, and found that a greater percentage of neighborhood parks could act as a buffer for reducing stress [31]. In a systematic review of the literature focused on children and adolescents, Moll et al. (2022) found an array of positive effects from interactions with and exposure to nature [40]. In the studies that focused specifically on adolescents and their engagement in natural habitats, findings included reduced stress levels, increased attention spans and working memory, and improved perceptions of mental health and self-satisfaction [41–43].

## Limitations

One limitation of using buffers for the park and NDVI data is that it represents distance "as the crow flies" and does not account for potential barriers to access (e.g., highways, fences, etc.). Additionally, we cannot account for actual park use or time spent in the park by students, only the proximity of the nearest park to the schools. However, as discussed previously, if we assume that students spend more time in public parks that are in closer proximity to their school, then our findings are consistent with the current literature on this topic. Despite this limitation, the magnitude of our results is notable given that there was not a structured intervention to compare students who spent time in natural habitats with those who did not, similar to other studies [20–22]. Additionally, the insignificance of the NDVI coefficients could be due to limited variation in the range of NDVI values in the data. Thus, these results may have limited generalizability to regions that have more diverse climates and greater variability in seasonal greenness. However, we do note that the mean and standard deviation of our NDVI values are in line with other studies that have examined the relationship between NDVI and academic performance and have similarly, found no statistically significant relationship [12] or a near zero, but negative and statistically significant relationship [18].

Another potential concern with the analysis would be if wealthier schools and school districts provided better education so students at these schools had higher test scores, and these schools were also more likely to be in close proximity to public parks. In this case, the park variable would not necessarily be capturing the impact of the park itself, but could be a proxy for school wealth and quality. To control for these confounding factors, we include a number of variables in the model to capture school quality, including teaching experience at the school level and per pupil funding and property tax

revenues at the school district level. Other variables to proxy for school wealth include a variable for the percent of low-income students at the school and a mean household income variable based on an 800 m radius around the school that is weighted by census tract. We also checked to see if there was a systematic relationship between the proximity to public parks and the general per pupil funding at the school district level. Specifically, we tested for correlation between the general per pupil funding and the presence of a park, as well as the percent of park within an 800m buffer area of the school for all five years. The highest correlation values were −0.13 and −0.041, respectively, suggesting little correlation between the variables. Moreover, the negative values suggest that wealthier school districts actually have less access to parks.

Another limitation of this study is data suppression issues pertaining to the student assessment results. The data measuring the academic outcomes was obtained from OSPI, which applies suppression rules to any aggregated student data that is connected to student academic outcomes. Therefore, the data on the annual percentage of students who met or exceeded proficiency in a given standardized test is suppressed if they have less than 10 total students in the grade. This would be an issue if schools that have suppressed data are systematically farther away from public parks due to their more rural status, but have higher percentages of students who meet or exceed standards on assessment tests on average, which would lead to an upward bias coefficient for the association of park proximity on educational outcomes in our dataset. To account for this, we include the rural binary variable in our analysis. Additionally, given that our analysis focuses on urban counties in Washington, data suppression is less likely to have a significant impact in our findings due to larger class sizes.

Finally, due to data limitations we are unable to include annual park data in our study. Therefore, measures such as presence and percent of park within the various buffer areas, and distance to the nearest park are calculated based on the most updated park shapefiles provided by the city or county of record. All of the park data has been updated within the time frame of our analysis and we assume that the number and size of the parks have not changed in a meaningful way given that the addition or expansion of parks typically fall under a county or city master plan and require longer time scales than the period of analysis in this study. That said, while it is unlikely that new parks were added into the study area, upgrades and park renovations may have occurred. However, since we do not account for the quality of a park in this study, only the proximity of the park to the school, changes to the park over time would not impact our results.

## Conclusion

We compare the impact of proximity to a public park and general greenness measured by NDVI on academic performance at the elementary, middle, and high school levels in four counties in western Washington state. Overall, we find that for middle schools, being within an 800 meter radius of a public park is associated with a 4.2 percent increase in the number of students that score proficient or higher on standardized tests on average. These results are stronger for students in grade 8 compared to grade 7, with positive and statistically significant results across all tests types – Math, Science, and Language Arts. The results were more mixed and not statistically significant at the elementary and high school levels. NDVI, as a measure of general greenness, did not have a relationship with test performance. Our findings also suggest that at the middle school level, student socioeconomic status is a stronger predictor of test performance than racial composition, though the percent of Asian students within a grade is consistently positive and statistically significant while Native Indian percentage is consistently negative and statistically significant.

The findings from this study aim to inform policy decisions about public park access in the state. When determining park placement, prioritizing locations near low-income middle schools will not only confer the benefits of parks to the local community members, but could also potentially boost school outcomes measured through standardized test scores. Overall, more research looking at the relationship of public parks and schooling outcomes is needed to further corroborate our results, but we believe that having more access to public parks could benefit students in their educational and academic performance in addition to the myriad other benefits associated with greater engagement in natural habitats.

## Supporting information

**S1 Table. Regression results by grade-level for 800 m buffer area and 800 m spring NDVI.**
(PDF)

**S2 Table. Comparison across 400 and 250m buffer Spring & 800m, 400m, and 250m Fall NDVI with 800m buffer measure of greenspace for middle school.**
(PDF)

## Acknowledgments

The authors would like to thank to Washington State departments of Natural Resources, Parks, and Office of Superintendent of Instruction for sharing their data and the two anonymous referees for helpful suggestions on data, analytical approach, presentation, and literature references.

## Author contributions

**Conceptualization:** Lea Fortmann, Isha Rajbhandari.

**Data curation:** Lea Fortmann, Isha Rajbhandari.

**Formal analysis:** Lea Fortmann, Isha Rajbhandari.

**Methodology:** Lea Fortmann.

**Writing – original draft:** Lea Fortmann, Isha Rajbhandari.

**Writing – review & editing:** Lea Fortmann, Isha Rajbhandari.

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
