## [Decision Letter · Decision Letter 0]

4 Mar 2025

PONE-D-24-39146Is All Greenspace Created Equal? Assessing the impact of public parks and greenness on academic achievement in Washington StatePLOS ONE?

Dear Dr. Fortmann,

Thank you for submitting your manuscript to PLOS ONE. After careful consideration, we feel that it has merit but does not fully meet PLOS ONE’s publication criteria as it currently stands. Therefore, we invite you to submit a revised version of the manuscript that addresses the points raised during the review process.

We look forward to receiving your revised manuscript.

Kind regards,

Giulia Squillacioti

Academic Editor

PLOS ONE

Journal Requirements:

2. In your Methods section, please include additional information about your dataset and ensure that you have included a statement specifying whether the collection and analysis method complied with the terms and conditions for the source of the data.

4. We note that Figure 1 in your submission contain map/satellite images which may be copyrighted. All PLOS content is published under the Creative Commons Attribution License (CC BY 4.0), which means that the manuscript, images, and Supporting Information files will be freely available online, and any third party is permitted to access, download, copy, distribute, and use these materials in any way, even commercially, with proper attribution. For these reasons, we cannot publish previously copyrighted maps or satellite images created using proprietary data, such as Google software (Google Maps, Street View, and Earth). For more information, see our copyright guidelines: http://journals.plos.org/plosone/s/licenses-and-copyright.

Reviewers' comments:

Reviewer's Responses to Questions

**Comments to the Author**

1. Is the manuscript technically sound, and do the data support the conclusions?

Reviewer #1: Yes

Reviewer #2: Partly

2. Has the statistical analysis been performed appropriately and rigorously?

Reviewer #1: Yes

Reviewer #2: No

3. Have the authors made all data underlying the findings in their manuscript fully available?

Reviewer #1: No

Reviewer #2: Yes

4. Is the manuscript presented in an intelligible fashion and written in standard English?

Reviewer #1: Yes

Reviewer #2: Yes

Reviewer #1: Thank you for the opportunity to review this manuscript which explores the association between school proximity to public parks and academic performance using fixed effects analysis with five-year panel data.

Overall, the manuscript makes an interesting contribution to the growing literature on access to parks, greenery and academic performance.

I have some general comments followed by some more specific ones.

The content of the manuscript requires some rearranging in line with the typical layout of a scientific manuscript. In particular the study results should not be described in under ‘I Introduction’ (as they are currently described on page 4). Much of the latter part of the ‘Introduction’ from the last paragraph on page 3 onwards belongs in the Discussion section towards the end of the manuscript.

I have now read that “PLOS ONE waives all formatting requirements until your manuscript has received a provisional Editorial Accept decision.” I still consider it more standard to have an Introduction that describes background, rationale and aims for the study; then a Methods section; followed by a Results section, separate from a Discussion section. Results should not be described in the Introduction.

The benefits of parks for children and adolescents are described throughout the manuscript. At some point the role of parks as a setting for being physically active should be described. Whilst the current study does not examine physical activity (PA) it is important to acknowledge that some beneficial associations between parks and academic performance may have pathways via PA. Positive associations have been shown between PA and academic performance, although not all findings are consistent (e.g. see Marques et al., 2017).

Marques A, Santos DA, Hillman CH, Sardinha LB. How does academic achievement relate to cardiorespiratory fitness, self-reported physical activity and objectively reported physical activity: a systematic review in children and adolescents aged 6–18 years. British journal of sports medicine. 2017;52(16):1039.

Abstract

Please include “, USA” after “western Washington state”.

Page 2-3

It would be good to include some studies from beyond the USA. For example, significant associations have been found between surrounding greenery and performance in national standardized school tests in Australia:

1. Carver A, Molina MA, Claesen JLA, Klabbers G, Donaire D, Gonzalez, et al. Vegetation and vehicle emissions around primary schools across urban Australia: associations with academic performance. Environmental Research. 2022:113256.

2. Claesen JLA, Wheeler AJ, Klabbers G, Gonzalez DD, Molina MA, Tham R, et al. Associations of traffic-related air pollution and greenery with academic outcomes among primary schoolchildren. Environmental research. 2021:111325.

II Data and Methods

Page 5, second paragraph

Please include references for county and city parks and recreation department websites.

“We also assess the overall greenness of the area surrounding the school at various buffer distances…”

Please describe the buffer distances.

Page 6, paragraph 2

“We also include a number of student and school controls in our analysis including race, gender, income, and teacher quality variables”.

Please replace “controls” with “covariates”.

How was “low-income” defined?

Please be consistent when describing the buffer radius, i.e. either 800 m or half a mile; please do not use them interchangeably.

Table 1 (should be in Results section)

Include values for NDVI, e.g. mean and standard deviation or median and range.

Table 2 (should be in Results section)

Rather than “(dummy)” after “Park 800m”, it would be clearer to write “Park within 800m”.

Last sentence of section III appears to be missing a final word:

“If parks are more likely to have greater tree cover, then these results coincide with our findings that tree covered parks have a stronger relationship with”.

Reviewer #2: I indicated that the statistical analysis was not performed rigorously because it appears that the authors want to make a causal inference, and I don't think that the methods support this. The methods may be rigorous enough if they are wanting to make different conclusions, and perhaps added additional control variables. More details below:

This paper examines the relation for greenspace around schools to academic achievement at the school level. The paper utilizes multiple metrics of greenspace to determine that schools that have a public park located within ½ mile radius have higher student achievement levels than do schools without a park in the ½ mile radius. This relation was found only among middle schools but not among elementary or high schools. Greenspace as measured by the NDVI was not associated with student achievement.

Overall, I found the paper to be interesting but felt that the description of the findings and conclusions were not completely in line with the analysis methodology used.

The authors use causal language describing ‘impact’ of greenspace throughout the paper. Based on my understanding of their methods, I don’t think that any causal conclusions can be drawn.

The inclusion of multiple measures of greenness and various control variables is a strength of the study as the authors discuss, but I wonder why other school neighborhood variables were not included. It seems that including these would be necessary to advancing the argument that parks, rather than neighborhood resources more generally, are related to achievement.

Associations between school racial/ethnic composition and school achievement are presented without discussion of systemic racism as a reason for these differences.

Analyses were done predicting school level aggregates of achievement, but findings are discussed in terms of how parks might impact individual children’s behavior. Is this a correct interpretation of results? Have the authors considered whether this may be an example of the ecological fallacy?

Additionally, I was surprised that the authors described the results of their study in the introduction. This may be a disciplinary difference.

.

Reviewer #1: No

Reviewer #2: No

---

## [Author Response · Author response to Decision Letter 1]

2 Sep 2025

All responses to the reviewer's and editors comments and included in the file "Response to Reviewers"

---

## [Decision Letter · Decision Letter 1]

22 Oct 2025

PONE-D-24-39146R1Is all greenspace created equal? Assessing the relationship of public parks and greenness on academic achievement in Washington, USAPLOS ONE?

Dear Dr. Fortmann,

We look forward to receiving your revised manuscript.

Kind regards,

Giulia Squillacioti

Academic Editor

PLOS ONE

Journal Requirements:

Reviewers' comments:

Reviewer's Responses to Questions

**Comments to the Author**

Reviewer #1: (No Response)

Reviewer #2: (No Response)

2. Is the manuscript technically sound, and do the data support the conclusions?

Reviewer #1: Yes

Reviewer #2: Yes

3. Has the statistical analysis been performed appropriately and rigorously?

Reviewer #1: Yes

Reviewer #2: I Don't Know

4. Have the authors made all data underlying the findings in their manuscript fully available?

Reviewer #1: Yes

Reviewer #2: Yes

5. Is the manuscript presented in an intelligible fashion and written in standard English?

Reviewer #1: Yes

Reviewer #2: Yes

Reviewer #1: Thank you for the opportunity to review this revised manuscript. Overall, the manuscript is much improved and I am s satisfied with the authors responses to my comments.

I have noticed a couple of points requiring attention before publication:

Line 246 I think this should say ‘Table 3’ rather than ‘Table 2’.

Line 255 Please check whether the percent level is correct - should it say ‘23’ instead of ‘10’?

Reviewer #2: The authors have done a nice job of responding to most of the questions raised in the first review of the manuscript. A few original comments may warrant further attention, and their revision has raised a few new questions.

Lines 113-114 . Unclear what is meant by “to account for more rural areas in western Washington counties, we include rural and urban school district data…”. Earlier the authors say they focus on western Washington state because of its urban nature.

In my first review, I had suggested including other neighborhood level characteristics as controls. The authors stated that this was not possible due to data limitations. I may not be fully understanding the limitations of the available data, but it still seems to me that it would be possible to use publicly available census data to capture neighborhood level socioeconomic characteristics within the buffers created for the NDVI by using weighted averages of census tract variables. I recognize, however, that the authors may have conceptual or other methodological reasons for not controlling other neighborhood level variables.

The authors state that there is not substantial variation in greenness among the schools. If there is not substantial variation in greenness among the schools, then is it meaningful to examine associations of greenness with academic performance? It seems problematic to conclude that NDVI is not related to academic performance in a study where there is limited variation in greenness among schools. This seems to be a major limitation. The authors do mention that this may also have implications for the generalizability of these findings; they may want to consider moving this information to the limitations section.

Results- It would be helpful to contextualize the percentages presented as effect sizes or in some other relevant way.

Assessment test outcomes are not well described in the materials and methods section. Why is the outcome dichotomous rather than a categorical measure of achievement that would capture more variation (i.e., separating out meets and exceeds expectations)?

Lines 241 - ‘seems to be more important’ – has a bigger impact in?

Lines 242 – ‘more significant’ – does this mean smaller p-value or larger magnitude of impact?

The results section includes discussion of the results as well. This information could be moved to the discussion section.

In the conclusion, the authors describe “a decline in academic performance”. Do they mean lower academic performance or a decline?. The authors also refer to ‘racial and socioeconomic classes’ ; that terminology seems problematic.

I agree that there does not need to be a full discussion paragraph on the potential drivers of the socioeconomic and racial disparities findings. However, as the racial disparities in test scores are highlighted throughout the manuscript, including in the conclusions section, I think that it is appropriate to add a phrase or sentence contextualizing or explaining those findings.

.

Reviewer #1: No

Reviewer #2: No

---

## [Author Response · Author response to Decision Letter 2]

30 Jan 2026

We have provided a detailed response to the Reviewer's comments in the documents provided.

---

## [Decision Letter · Decision Letter 2]

16 Mar 2026

PONE-D-24-39146R2Is all greenspace created equal? Assessing the relationship of public parks and greenness on academic achievement in Washington, USAPLOS One?

Dear Dr. Fortmann,

Thank you for submitting your manuscript to PLOS ONE. After careful consideration, we feel that it has merit but does not fully meet PLOS ONE’s publication criteria as it currently stands. Therefore, we invite you to submit a revised version of the manuscript that addresses the points raised during the review process.

We look forward to receiving your revised manuscript.

Kind regards,

Giulia Squillacioti

Academic Editor

PLOS One

Journal Requirements:

Additional Editor Comments:

Please address the comments raised by the Reviewer #3

Reviewer's Responses to Questions

**Comments to the Author**

Reviewer #3: (No Response)

2. Is the manuscript technically sound, and do the data support the conclusions?

Reviewer #3: Yes

3. Has the statistical analysis been performed appropriately and rigorously?

Reviewer #3: Yes

4. Have the authors made all data underlying the findings in their manuscript fully available?

Reviewer #3: Yes

5. Is the manuscript presented in an intelligible fashion and written in standard English?

Reviewer #3: Yes

Reviewer #3: The authors have effectively addressed reviewer responses. I did note a couple of issues that should be corrected before publication and I raise one substantive issue that I ask the authors to consider:

Pg 6, referring to line 116 - there is a comment in the margins that says "While the study area includes only urban counties, our sample includes both rural and urban school districts from the National Center for Educational Statistics (NCES) [23]. Thus, our findings pertain to access to park in both rural and urban areas." That seems to be a more clear way of articulating this point and responding to the reviewer's concern, with perhaps a simple edit - "While the study area includes only urban counties, there are both urban and rural school districts (by definition of ...(NCES) in these counties. Thus, our findings pertain to access to parks in both rural and urban areas.

Line 146 Table 1 - this isn't a big deal, but it would improve clarity and avoid confusion if the table were divided into two sections (upper for data that are numbers with a relevant column heading - not means and SDs, and lower with the means and SDs heading for data relevant to representing means and SDs. )

Line 224 - "within a half mile" is a different unit of measurement than 800 meters, but it is a helpful one for audiences that think in miles not km. Perhaps just parenthetically noting the equivalence, like "...having a public park within a half mile radius (800 meters) of a school...."

Line 316 - I believe the authors missed changing this to 4.2

Line 373 - 376 - by saying "the latter" it seems you are saying that your results represented a negative relationship as opposed to no relationship. The use of the dichotomous former/latter way of speaking doesn't work when you want to refer to something that is the middle and third option.

Limitations section - A commonly discussed limitation of the use of NDVI in buffers is that it represents data "as the crow flies" and does not account for potential lack of access to that green space by, for example a highway, fencing, etc. Driving/walking routes to that greenspace are often used to address this. I'm not suggesting that data be re-analyzed, but it would be helpful to note in the Limitations section that NDVI "as the crow flies" does have its drawbacks.

Discussion- I understand that NDVI was a control for this study, not the focus of the analysis with outcome data, and that NDVI data was not disparate from that documented in past studies. But, it would help to note in the discussion that your finding of no NDVI/outcome relationship is in contrast to some previous findings, even those studies with similar amount of greenness, but overall is consistent with the mixed results in the literature.

This is my substantive suggestion for consideration:

Bottom of pg 12 - there is increasing interest in testing the equigenic effect - when children from lower SES backgrounds (as measures by family, school or community economic data) benefit RELATIVELY MORE from nature than children from higher SES backgrounds. It seems that lines 240 - 242 only tell part of the important story. Is it possible to analyze the middle school data stratified by a relevant SES variable, rather than simply controlling for it?

.

Reviewer #3: **Yes:** Catherine JordanCatherine JordanCatherine JordanCatherine Jordan

---

## [Author Response · Author response to Decision Letter 3]

24 Mar 2026

Response to Reviewer #3’s Comments

Reviewer #3: The authors have effectively addressed reviewer responses. I did note a couple of issues that should be corrected before publication and I raise one substantive issue that I ask the authors to consider:

Pg 6, referring to line 116 - there is a comment in the margins that says "While the study area includes only urban counties, our sample includes both rural and urban school districts from the National Center for Educational Statistics (NCES) [23]. Thus, our findings pertain to access to park in both rural and urban areas." That seems to be a more clear way of articulating this point and responding to the reviewer's concern, with perhaps a simple edit - "While the study area includes only urban counties, there are both urban and rural school districts (by definition of ...(NCES) in these counties. Thus, our findings pertain to access to parks in both rural and urban areas.

We thought we had deleted this comment, which was from a previous revision that we already addressed. However, we have now moved this information to the School Data section [line 125-127] since it describes specific school characteristics, which we think is a better fit and included the sentence:

“To account for schools located in more rural areas in western Washington, we include data from the National Center for Educational Statistics (NCES) that classifies schools as rural based on their distance from an urban area [23]. ”

Line 146 Table 1 - this isn't a big deal, but it would improve clarity and avoid confusion if the table were divided into two sections (upper for data that are numbers with a relevant column heading - not means and SDs, and lower with the means and SDs heading for data relevant to representing means and SDs. )

We have rearranged the table according to your suggestions, which we agree, makes more sense.

Line 224 - "within a half mile" is a different unit of measurement than 800 meters, but it is a helpful one for audiences that think in miles not km. Perhaps just parenthetically noting the equivalence, like "...having a public park within a half mile radius (800 meters) of a school...."

We have made this change in the manuscript (line 226).

Line 316 - I believe the authors missed changing this to 4.2

Thank you, we have made this change (line 311).

Line 373 - 376 - by saying "the latter" it seems you are saying that your results represented a negative relationship as opposed to no relationship. The use of the dichotomous former/latter way of speaking doesn't work when you want to refer to something that is the middle and third option.

We have changed the language here to take out the word “latter” (line 434).

Limitations section - A commonly discussed limitation of the use of NDVI in buffers is that it represents data "as the crow flies" and does not account for potential lack of access to that green space by, for example a highway, fencing, etc. Driving/walking routes to that greenspace are often used to address this. I'm not suggesting that data be re-analyzed, but it would be helpful to note in the Limitations section that NDVI "as the crow flies" does have its drawbacks.

We have added a line [388-390] to account for this limitation:

“One limitation of using buffers for the park and NDVI data is that it represents distance “as the crow flies” and does not account for potential barriers to access (e.g. highways, fences, etc.).”

Discussion- I understand that NDVI was a control for this study, not the focus of the analysis with outcome data, and that NDVI data was not disparate from that documented in past studies. But, it would help to note in the discussion that your finding of no NDVI/outcome relationship is in contrast to some previous findings, even those studies with similar amount of greenness, but overall is consistent with the mixed results in the literature.

The variation in NDVI in our study is consistent with other studies, specifically Kuo et al. (2021) and Browning, et al. (2018), that also found no statistically significant relationship between NDVI and academic performance in for former, and a small negative relationship in the latter. We briefly mention this in the Discussion section, but we have more discussion on it in the Limitations section, as stated below in starting in line 406.

“we do note that the mean and standard deviation of our NDVI values are in line with other studies that have examined the relationship between NDVI and academic performance and have similarly, found no statistically significant relationship [12] or a near zero, but negative and statistically significant relationship [18].”

This is my substantive suggestion for consideration:

Bottom of pg 12 - there is increasing interest in testing the equigenic effect - when children from lower SES backgrounds (as measures by family, school or community economic data) benefit RELATIVELY MORE from nature than children from higher SES backgrounds. It seems that lines 240 - 242 only tell part of the important story. Is it possible to analyze the middle school data stratified by a relevant SES variable, rather than simply controlling for it?

We analyzed the data stratified by the percent of low income students in middle schools, looking at the bottom poorest 25% of schools (60% or more of the students are reported as low-income) and the wealthiest 25% of schools (less than 20% of students are reported as low-income). The coefficient for the low-income schools is negative and the coefficient for the wealthiest schools is positive, though neither are statistically significant. For the middle 50% of schools, the coefficient was significant and similar in magnitude to the main results from Table 3 in the paper. We have compiled the results in a table (see the Word document with response to reviewer comments) but we have not added this table to the paper.

Ultimately, we feel that these results are too inconclusive to report and are outside the scope of our current paper, though we think that this is an important question worth exploring further in a more robust way and would like to do so in the future. Ideally we would like to have more data on park quality and potentially other neighborhood characteristics as additional control variables to have a more robust model for understanding how the relationship between parks and academic performance might vary by SES. For example, it may be the case that parks near lower income schools are of poorer quality and/or may be more susceptible to crime or homelessness, which could be driving these preliminary results.

---

## [Editor Report · Decision Letter 3]

31 Mar 2026

Is all greenspace created equal? Assessing the relationship of public parks and greenness on academic achievement in Washington, USA

PONE-D-24-39146R3

Dear Dr. Fortmann,

We’re pleased to inform you that your manuscript has been judged scientifically suitable for publication and will be formally accepted for publication once it meets all outstanding technical requirements.

Kind regards,

Giulia Squillacioti

Academic Editor

PLOS One
---

## [Editor Report · Acceptance letter]

PONE-D-24-39146R3

PLOS One

Dear Dr. Fortmann,

I'm pleased to inform you that your manuscript has been deemed suitable for publication in PLOS One. Congratulations! Your manuscript is now being handed over to our production team.

Kind regards,

on behalf of

Dr. Giulia Squillacioti

Academic Editor

PLOS One